# Neural signatures associated with temporal compression in the verbal retelling of past events

Elizabeth Musz [1✉] & Janice Chen [1]

When we retell our past experiences, we aim to reproduce some version of the original events; this reproduced version is often temporally compressed relative to the original. However, it is currently unclear how this compression manifests in brain activity. One possibility is that a compressed retrieved memory manifests as a neural pattern which is more dissimilar to the original, relative to a more detailed or vivid memory. However, we argue that measuring raw dissimilarity alone is insufficient, as it confuses a variety of interesting and uninteresting changes. To address this problem, we examine brain pattern changes that are consistent across people. We show that temporal compression in individuals' retelling of past events predicts systematic encoding-to-recall transformations in several higher associative regions. These findings elucidate how neural representations are not simply reactivated, but can also be transformed due to temporal compression during a universal form of human memory expression: verbal retelling.

[1] Department of Psychological and Brain Sciences, Johns Hopkins University, Baltimore, MD 21218, USA. ✉email: emusz1@jhu.edu

More than any other creature, humans have a great ability to describe their past experiences to others. When we retell past experiences from memory, the goal is to reproduce some version of the original events; this reproduced version is often shorter, sparser, or otherwise reduced from the original[1–5]. For example, when you tell a friend about a book you recently read, your retelling is unlikely to take as long as it took you to read the book. Instead, a temporally compressed summary, in which you have extracted what you consider to be the most important aspects of the book—the main plotline, the characters' personalities, the distinctive writing style—will be much more appreciated by your friend than a verbatim recreation.

The compression of experience into memory often occurs in a similar manner across people. Psychological studies show that, given the same events, different people tend to remember and forget in patterns common with each other. One source of commonality in remembering arises from the intrinsic memorability of items: if people are asked to remember a large number of images, some images are less likely to be forgotten than others due to factors such as distinctiveness and social content[6–12]. In memories of more complex material such as events in a narrative or autobiographical experience, the relative prominence of event features may be modified, such that some aspects are strengthened in the remembered version while other details are minimized[13–18]. Individuals also draw on their world knowledge when recalling their experiences[13,19]; over time, errors may be introduced that shift memory closer to existing schema[1,20], and recollections often become more semanticized, gist-like versions of the original events[21–25]. Such losses and changes between encoding and recall should not necessarily be viewed as failures; rather, if the changes are consistent across individuals, they may confer benefits such as building common ground needed for effective communication[26].

How does the compressed nature of memories manifest in brain activity? As with behavior, the retrieved or reconstructed neural representation is an imperfect copy of the original representation[27,28]. Many studies have shown that, when people retrieve episodic memories, brain activity patterns similar to those present at the time of encoding a given item or event are reinstated in a set of brain regions which typically include default mode network (DMN) and high-level sensory areas, with reinstatement strength modulated by retrieval strength or vividness[29–36]. Unlike vivid memories that allow us to re-experience the past, "compression" suggests the initially encoded events have been altered. Thus, a compressed memory might present as a neural pattern during retrieval which is more dissimilar to the original pattern relative to a more detailed or vivid memory. However, retrieval conditions may differ from the encoding conditions in many ways, such as the surroundings, the task demands, and the physical state of the rememberer. Given these caveats, a raw dissimilarity measurement alone between neural patterns at encoding and those during recall seems insufficient, as it would incorporate both: (1) interesting changes, e.g., behaviorally-relevant shifts in representations of specific items or events; and (2) uninteresting changes, e.g., broad differences between brain states when one is watching video versus producing speech, as well as random decay of memory traces[37]. Furthermore, while raw dissimilarity captures both subject-consistent and subject-idiosyncratic changes, it cannot discriminate subject-idiosyncratic changes from task-irrelevant noise.

A complementary approach would be a test which selectively measures neural changes between encoding and recall that are consistent across people. In a prior study, Chen et al. (2017)[2] observed that activity patterns in certain brain regions changed systematically from their form during encoding—in this case, watching a movie—to an altered form during spoken recollection of the movie. Importantly, the analyses showed that whatever activity pattern change occurred between watching a given movie event (the original) and recalling it (the reproduction), the change was *common* across brains yet *specific* to individual movie events. This approach selectively identifies subject-consistent changes, separating them from state-related changes, subject-idiosyncratic changes, and task-irrelevant noise[38]. While raw dissimilarity between encoding and recall patterns would increase with any change, an approach which computes only subject-consistent changes necessarily rules out random decay of memory traces, i.e., identified changes are consistent across brains and thus non-random. These cross-subject, systematic pattern changes were observed in DMN cortical regions, in keeping with observations that these regions participate in episodic memory retrieval[33] and carry information about mental models of situations or perspectives[39,40]. However, in this prior work, the neural effect was not grounded in a psychological description, e.g., not linked to any specific features of the movie contents, nor to the nature of the individuals' recollections[2].

In the present study, we hypothesized that subject-consistent changes in neural patterns between encoding and retrieval—changes that are common across brains yet specific to individual events—would be related to the compression of events as participants retold them from memory. Compression, or summarization, was assessed in the spoken recall of a recently viewed audiovisual movie by quantifying, for any given utterance of verbal recollection, the duration of the described event during encoding (i.e., movie-viewing). Behaviorally, we observed that speakers naturally varied the level of temporal precision that they provided for any given event. Some recollections pinpointed a specific moment (e.g., "I picked up the knife and sliced the butter"), while other descriptions summarized over longer periods of time (e.g., "I ate breakfast"), i.e., temporally compressed events to a greater degree. In neural analyses, we first asked whether summarization during recall was associated with greater raw pattern dissimilarity between encoding and recall; multi-voxel correlation analyses showed that, relative to temporally precise utterances, summary utterances during recall were indeed associated with reduced neural reinstatement in the posterior medial cortex, a key DMN region, as well as weaker reductions in other posterior DMN regions. Critically, we next examined changes that were *common* across brains yet *specific* to individual movie events. As predicted, we observed that summarization was significantly positively associated with encoding-to-recall changes in a number of high-level associative and visual areas: events that were later summarized underwent a greater transformation of their neural patterns between initial perception and later recall, relative to events described with high temporal precision. Our findings show that reactivation effects, typically observed in the DMN, are modulated by the degree to which people temporally compress recollected naturalistic events.

## Results

**Description of the paradigm and dataset.** Participants ($n = 17$) viewed an audiovisual stimulus, the first 50 min of Episode 1 of BBC's *Sherlock*, while undergoing fMRI. All participants reported that they had not seen any episodes of the series before. Participants were informed that immediately after viewing the video, they would be instructed to describe what they had seen. After the end of the video, participants verbally recalled the plot of the movie aloud in as much detail as possible and in their own words, also during fMRI scanning (Fig. 1a). The speech was recorded with a microphone. No visual or auditory cues were provided during the recall session. Data were from Chen et al. (2017)[2,41].

**Classification of summarization vs. temporal precision for individual utterances during spoken recall.** To characterize summarization during recall, we created a rubric to score

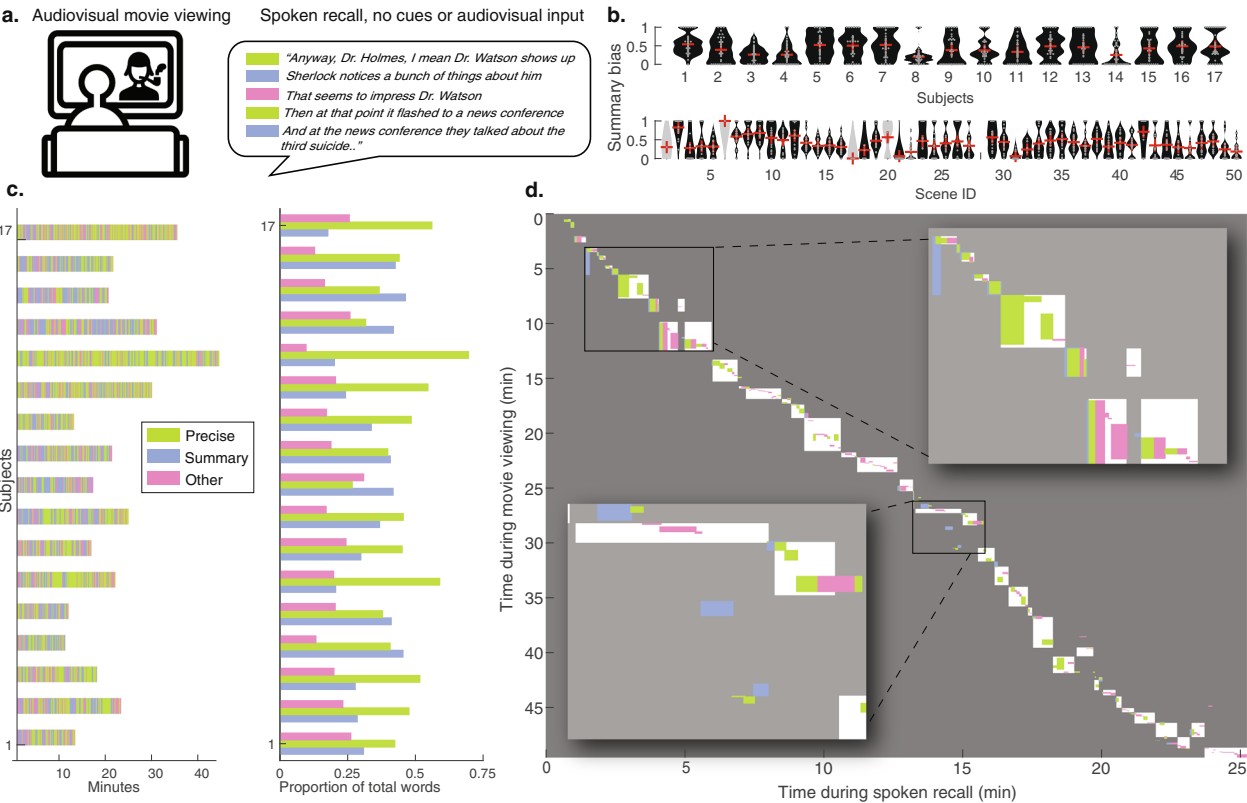

**Fig. 1 Experiment design and recall behavior. a** Each subject participated in two fMRI scanning runs. During Run 1, participants watched a 50-min movie, BBC's *Sherlock*, Episode 1. During the immediately following Run 2, participants recalled the movie content via verbal recall. Participants' responses were audio-recorded, transcribed, and segmented into utterances that roughly aligned with the ends of sentences and breaks in speech (each row). Each utterance was categorized according to the movie content that it referred to, and according to its temporal precision (i.e., the amount of time that it took for the described events to elapse in the movie). Utterances that described events that occurred in 10 s or less were coded as Temporally Precise, while events that spanned longer than 10 s were coded as Summary. Utterances were coded as Other if they did not provide literal and accurate descriptions of the movie events. **b** Violin plots depicting the Summary bias across scenes for each individual subject (top row) and across subjects for each individual scene (bottom). Scenes shown in gray were recalled by less than 5/17 participants and were excluded from all subsequent inter-subject analyses. **c** Subject-specific timelines of recall utterances, colored by the degree of temporal precision, and proportion of spoken words during recall that were included in utterances coded as Summary, Precise, or Other. **d** Diagram of scene durations during movie viewing (y-axis) and movie recall (x-axis) and scene order during recall (diagonal) in a representative participant. Each white box shows one scene from the original 50-scene segmentation[2]. Overlaid on top are the durations and scene identities of the specific recall utterances, coded by temporal precision. The insets show zoomed-in subsets of this recall behavior. For illustration purposes only, Other recall utterances are labeled as the most recently described movie scene. The images in Fig. 1a were purchased from The Noun Project, https://thenounproject.com.

individual utterances spoken by participants during their verbal recall of the movie. The rubric assesses the degree to which a given utterance refers to a specific moment of the movie (Temporally Precise), as opposed to describing a longer period of time (Summary). Here, utterances correspond roughly to sentences; in continuous speech, it is often not clear how to delineate sentences, and thus transcriptionists were instructed to break the text into short sentences using their best judgment. Each participant's speech was labeled sentence-by-sentence by two coders. Summarization was calculated in three ways: Direct judgment, Automatic, and Temporal compression factor. As the three yielded similar results, in the remainder of the paper we report results derived from the Direct judgment procedure: utterances were classified as Temporally Precise if they described movie events that elapsed in less than ten seconds and classified as Summary if they described events spanning more than ten seconds during the movie. See Methods for further details about the rubric and comparisons between all three approaches.

The total duration of the recall session varied across participants, as did the total number of recall utterances, ranging from 114 to 468 utterances ($M = 203.1$, SD = 110.1).

Each utterance was comprised of 14 words on average (SD = 2.4), and the mean utterance duration was 6.83 s (SD = 1.6 s). On average, 46.7% of participants' utterances were coded as Temporally Precise and 29.8% as Summary. Across participants, an average of 23.6% of utterances were labeled as Other (statements that were inaccurate or not about specific movie events) and excluded from subsequent analyses (Fig. 1). To compare the relative amounts of the different speech categories present in each subject's recall, we counted the number of words included in each category, summing across all of a subject's statements (Fig. 1c). Most participants produced a greater proportion of Temporally Precise versus Summary words (11/17 participants), although the number of words and the relative proportion of utterances in the different categories varied across individuals. For additional descriptive statistics see Supplementary Table 1.

**Summary utterances diverge from the original movie contents: a text-based analysis.** Summarization entails modifying the original event content, e.g., by synthesizing multiple elements of, or

selectively recalling only some aspects of, the original experience. As a validation test for our rubric, we sought to verify that recall utterances classified as Summary diverged more from the original material than those classified as Temporally Precise. To this end, we performed a text-based analysis using natural language processing tools to estimate the similarity between movie events and recall utterances.

The movie content was transcribed and annotated; then, these detailed natural language descriptions of the movie events were segmented into 1000 unique micro-segments, each approximately 4 s in duration (SD = 2.2 s, min = 1 s, max = 19 s), and then the recall utterances for each subject were linked to the matching movie micro-segment from a written description of the movie (see for details). After converting each of the movie micro-segment annotations and recall utterances into a unique 512-element sentence embedding vector using the Universal Sentence Encoder[42], we computed similarity between each recall utterance vector and all movie segment vectors for each participant. Movie vs. recall text cosine similarity was reliably greater for matching versus mismatching movie micro-segments and recall utterances; this was true for both Summary utterances, $t(16) = 32.54$, $p = 4.7e-16$, $d = 7.89$, 95% CI [0.10, 0.11] ($M$ cosine difference = 0.10, SD = 0.01) and Temporally Precise utterances, $t(16) = 44.77$, $p = 3.1e-18$, $d = 10.86$, 95% CI [0.18, 0.19] ($M$ cosine difference = 0.18, SD = 0.02). Critically, movie vs. recall text cosine similarity was significantly lower for micro-segments that were later summarized, relative to micro-segments that were precisely recalled, $t(16) = 10.26$, $p = 1.9e-8$, $d = 2.49$, 95% CI [0.06, 0.09] ($M$ cosine difference = 0.07, SD = 0.04); this effect was in the same direction for all individual subjects (17 of 17) (Supplementary Fig. 1). These results provide support for the effectiveness of our rubric in separating utterances which compressed across longer periods of movie time (Summary) from those which described events Temporally Precise manner.

To what extent did Summary utterances and Temporally Precise utterances arise from a recall of distinct events? In other words, did a participant tend to recall one set of events in a summarized manner and a different set of events in a temporally precise manner? For each subject, each Summary utterance's micro-segments were compared against the micro-segments associated with each Temporally Precise utterance. On average, 30% of a subject's Summary utterances covered micro-segments that were also described during Temporally Precise utterances (SD = 17.3, min = 7.7%, max = 74%).

**Encoding-Recall neural pattern similarity is weaker for summarized than for temporally precise utterances**. We next examined the strength of neural reactivation, i.e., raw pattern similarity, associated with Summary utterances as opposed to Temporally Precise utterances. By design, Summary utterances are associated with greater temporal compression of the memory relative to the encoded material; and as revealed above in the text-based analysis, Summary utterances diverge more from the original movie contents in terms of sentence-level semantics. Thus, we predicted that Summary utterances should be accompanied by weaker reactivation (greater raw pattern dissimilarity) than Temporally Precise utterances.

To measure memory reactivation, we computed the correlation between a given subject's brain pattern while (1) viewing specific moments of the movie versus (2) recalling the same moments. In order to separate the conditions of Summary and Temporally Precise to the greatest possible extent, this analysis was performed at the micro-segment level, i.e., separately for movie micro-segments that were later summarized versus those that were later described in a temporally precise manner.

The reinstatement analysis (movie vs. recall pattern correlation) was conducted first in a posterior medial cortex (PMC) ROI motivated by the previous study[2], and then in cortical parcels across the whole brain (citations and see Methods). Reinstatement was significant in the PMC ROI both for movie micro-segments that were recalled in a Temporally Precise manner, $t(16) = 9.49$, $p = 5.6e-8$, $d = 2.30$, 95% CI [0.08, 0.12] (mean $r = 0.10$, SD = 0.04) and those that were later Summarized, $t(16) = 7.07$, $p = 2.7e-6$, $d = 1.71$, 95% CI [0.04, 0.07] (mean $r = .05$, SD = 0.03). Furthermore, in 16/17 subjects, reinstatement was greater for Temporally Precise than for Summarized recall; $t(16) = 5.11$, $p = 0.0001$, $d = 1.24$, 95% CI [0.03, 0.07], all two-tailed tests (Fig. 2a). In the whole-brain parcel-based analysis, several regions exhibited significant reinstatement for both the Temporally Precise (Fig. 2b) and Summarized (Fig. 2c) micro-segments, including multiple parcels in lateral prefrontal and lateral temporal cortex, as well as posterior medial areas. The plurality of regions exhibiting these effects were located in parcels within the DMN (as defined from the parcellation atlas; see Fig. 3b), including 36% of the parcels in the Temporally Precise reinstatement map, and 30% in the Summarized reinstatement map (Table 1). These results are consistent with reinstatement effects reported in the searchlight analysis of the same data[2] as well as with many observations in the literature using a variety of stimuli and retrieval methods[29–31].

Several parcels in the DMN showed numerically greater reactivation for Temporally Precisely recalled versus Summarized movie micro-segments, including the left lateral parietal cortex and posterior medial cortex (Fig. 2d); however, for this contrast, no parcels survived FDR correction at $q = 0.05$. See Fig. 3b and Supplementary Table 2 for lists of the parcels defined as DMN and PMC, respectively.

In sum, Summary utterances during recall were associated with weaker neural reinstatement in PMC, relative to Temporally Precise statements, and subsequent brain-wide analysis found a number of DMN parcels in the medial and lateral parietal cortex trending in the same direction. These results suggest that temporal compression during recall modulates reinstatement effects in DMN subregions. Whereas previous work has reported that DMN reinstatement can vary depending on subjective ratings of memory vividness[34,36], here we show that the temporal compression of movie events, as identified in individuals' naturally varying speech content during spoken recall, is associated with decreased pattern similarity between encoding and recall.

**Regions exhibiting the memory transformation effect**. We hypothesized that subject-consistent changes in neural patterns between encoding and retrieval—changes that are common across brains yet specific to individual events—would be related to the behaviorally-identified compression of retrieved events. Thus, we next identified brain parcels that exhibited cross-subject-consistent changes between encoding and retrieval, i.e., a transformation effect. This analysis reiterates an analysis conducted by Chen et al. (2017)[2], now using a parcel-based approach rather than a searchlight procedure.

Chen et al. (2017)[2] introduced a simple method for identifying functionally relevant changes between encoding and recall patterns by leveraging similarity between individuals. In brief, the reasoning was that if memory-evoked patterns are more similar between individuals than to the original event pattern, then it can be inferred that patterns were transformed between perception and memory in a systematic (non-noise) manner. Using this method, the encoding-to-recall transformation was observed in many subregions of the default mode network in a searchlight analysis.

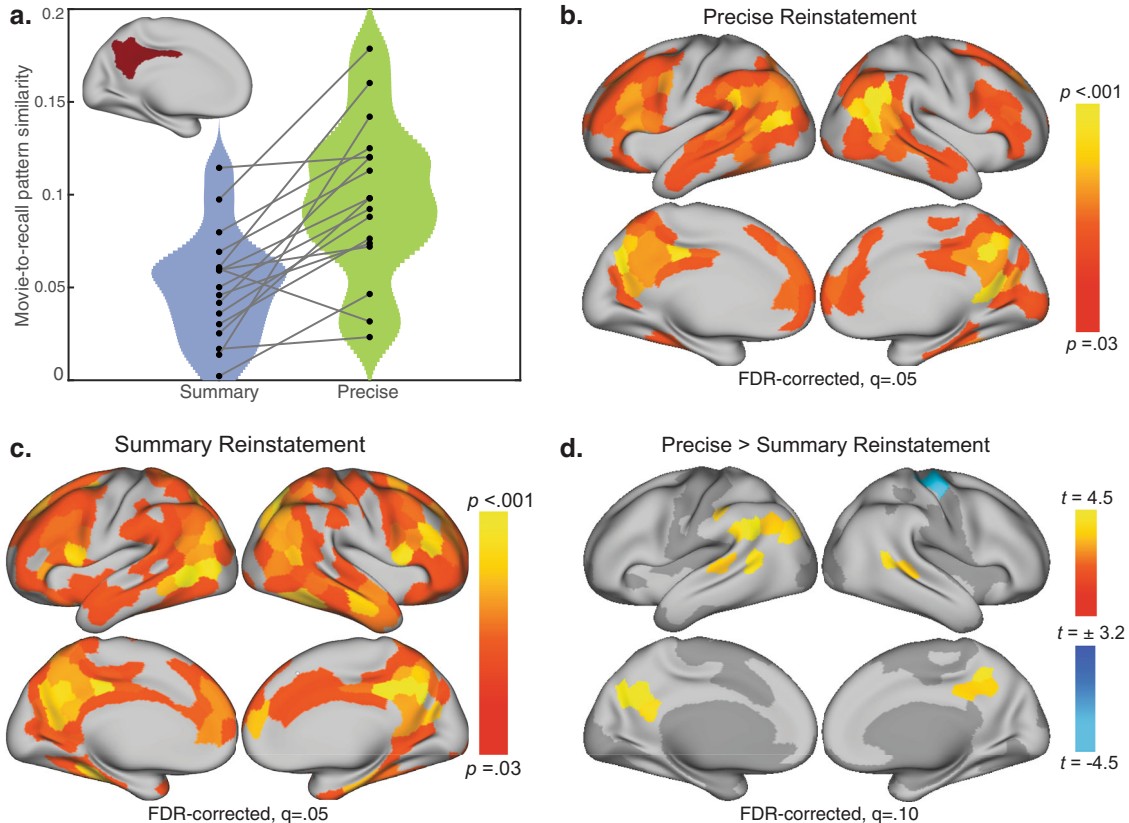

**Fig. 2 Within-subject movie-recall reinstatement analysis for movie micro-segments later recalled in a Summarized versus Temporally Precise manner.** Pattern similarity was performed at the individual-subjects level using movie micro-segments of short duration ($M = 4$ s, $SD = 2.2$ s). **a** Reinstatement analysis in the bilateral PMC ROI. Movie-recall pattern similarity was greater for movie micro-segments recalled in a Temporally Precise as opposed to a Summarized manner, $t(16) = 5.11$, $p < 0.001$, two-tailed. **b** Parcel map of the movie versus recall pattern similarity, limited to movie segments that were later recalled in a Temporally Precise manner. **c** Parcel map of movie-recall pattern similarity, limited to movie micro-segments that were later Summarized during recall. **d** Parcels where movie-recall similarity was greater for Temporally Precisely recalled segments versus Summarized segments. This analysis was limited to parcels that showed reliable reinstatement effects in either the Temporally Precise (2b) or Summary (2c) reinstatement maps; excluded parcels are shown in dark gray.

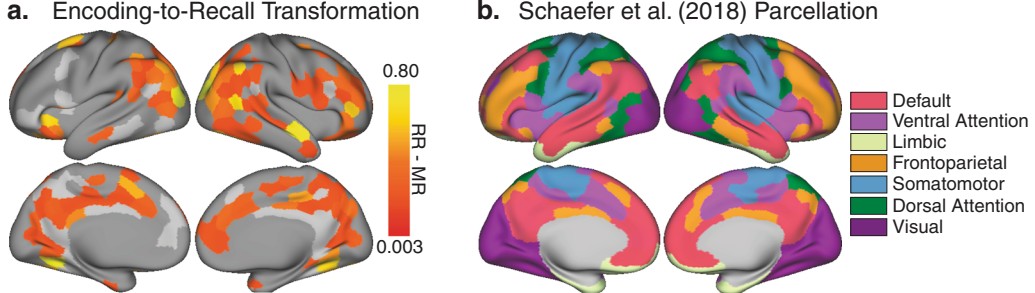

**Fig. 3 Parcel maps of encoding-to-recall memory transformation. a** Parcel map depicting group-level memory transformation effects, where scene-level recall-to-recall similarity (RR) was greater than movie-to-recall similarity (MR) across subjects (136/174 parcels). Values were only computed in parcels within a reliability mask, which was limited to parcels where either RR similarity or MR similarity was reliable (174/400 parcels). Parcels outside of this reliability mask are shown in dark gray, and parcels in the reliability mask where RR did not exceed MR are shown in light gray. Subsequent analyses were limited to parcels that exhibited memory transformation (all colored parcels). **b** All reported similarity values were computed in 400 parcels from an independent whole-brain resting-state parcellation[43]. Each parcel is associated with one of seven functional networks[43,82].

For the present study, we employed a parcel-based approach, and thus our first step was to recalculate transformation within individual parcels (400 parcels[43]). We first identified and averaged across the timepoints (TRs) that corresponded to each of the 50 movie scenes for movie-viewing and for recall, and then computed the parcel maps for movie-to-recall (MR) similarity and recall-to-recall (RR) similarity across people. The brain was then masked to retain only parcels that exhibited reliable activity patterns across subjects during movie-viewing or recall (174/400 parcels, see Methods). Within this mask, transformation scores were computed by subtracting each parcel's movie-recall correlation from its recall–recall correlation value. Transformation scores were positive in 136 of 174 parcels and these were retained for subsequent analyses (Fig. 3a). These parcels were distributed throughout several networks, including DMN (29% of parcels), the Visual Network (21% of parcels), the Frontoparietal Network

**Table 1 Quantity and location of brain parcels that show encoding-to-recall pattern similarity.**

| Parcel map | Relevant figure | Total parcels | Default | Fronto-temporal | Visual | Ventral attention | Dorsal attention | Somato-motor | Limbic |
|---|---|---|---|---|---|---|---|---|---|
| 1. Precise reactivation | Fig. 2b | 207 | 0.36 (75) | 0.18 (38) | 0.15 (30) | 0.14 (29) | 0.12 (24) | 0.03 (7) | 0.02 (4) |
| 2. Summary reactivation | Fig. 2c | 243 | 0.30 (73) | 0.19 (45) | 0.13 (32) | 0.16 (39) | 0.10 (25) | 0.08 (20) | 0.04 (9) |
| 3. Precise > Summary reactivation | Fig. 2d | 14 | 0.71 (10) | 0 | 0 | 0.07 (1) | 0.21 (3) | 0 | 0 |
| 4. Summary > Precise reactivation | Fig. 2d | 2 | 0 | 0 | 0 | 0 | 0 | 1 (2) | 0 |
| 5. Memory Transformation | Fig. 3a | 136 | 0.29 (39) | 0.15 (21) | 0.21 (28) | 0.13 (18) | 0.11 (15) | 0.07 (9) | 0.04 (6) |
| 6. Summary bias and memory transformation | Fig. 4b | 41 | 0.27 (11) | 0.20 (8) | 0.22 (9) | 0.20 (8) | 0.07 (3) | 0.05 (2) | 0 |

For each map, above-threshold parcels were distributed across the seven functional networks identified in Yeo et al., 2011 (see Fig. 3b). Each row lists the proportion (and count) of parcels located in each network. Analyses corresponding to rows 1–2 were tested in all 400 parcels. Rows 3–4 were limited to parcels that showed reliable effects in either row 1 or row 2 (283 total parcels tested); row 5 was limited to parcels where either encoding-to-recall or recall-to-recall was reliable (174 total parcels); and row 6 was limited to parcels that showed reliable effects in row 5 (136 total parcels). Results were thresholded at the $q < 0.05$ level, except for rows 3–4 ($q < 0.10$).

(15% of parcels), and the Dorsal (13%) and Ventral (11%) Attention Networks (see Fig. 3b and Table 1). Neither anterior hippocampus nor posterior hippocampus showed memory transformation or reliable within-subject reinstatement effects (Supplementary Note 1). A comparison to the searchlight-based transformation map from Chen et al. (2017)[2] is provided in Supplementary Fig. 2. For estimates of test-retest reliability for recall-to-recall similarity, movie-to-recall similarity, and memory transformation, see Supplementary Fig. 3.

**Summarization during recall scales positively with memory transformation**. We next sought to test our hypothesis that compression, as identified by summarization behavior in recollection speech, would be positively related to transformation—the magnitude of cross-subject-consistent pattern change from encoding to subsequent recall. We first computed scene-level Summary bias scores for all participants, as the transformation was calculated at the 50-scene level in the prior study (in contrast to the 1000 micro-segment levels we used in the reinstatement analysis reported above in Fig. 2). Summary bias was computed, for each movie scene, as the proportion of words describing that scene during recall that were from Summary utterances, versus Temporally Precise utterances. While the movie scenes varied in duration ($M = 37.3$ s, SD = 27.8), scene duration was not correlated with Summary bias across participants, $t(16) = -0.3$, $p = 0.76$ (mean $r = -0.01$, SD = 0.16). Participants varied in the degree to which they summarized each movie scene (Fig. 1b). Across movie scenes, the relative prominence of summarization does not appear to systematically differ depending on the movie timeline, although most participants summarize the second movie scene and produce temporally precise utterances about the final scene (Fig. 1b).

To test whether a bias toward summarization was positively related to memory transformation across movie scenes, we computed the correlation between these variables in each brain parcel. Testing was constrained to parcels that exhibited positive memory transformation scores in the prior analysis (i.e., parcels where recall–recall similarity exceeded movie-recall similarity; see Fig. 3a), as our interpretation of negative memory transformation scores is that subject-consistent encoding-to-recall changes have not been detected. We found a positive relationship between Summary bias and memory transformation in a number of parcels, such that scenes with a greater Summary bias underwent greater pattern changes from movie to recall, relative to scenes with a greater bias toward being recalled in a Temporally Precise manner (Fig. 4b): 40 parcels passed FDR correction at $q = 0.05$. These parcels were distributed across several functional networks: 27% of parcels were located in the DMN, 22% in the Visual Network, and 20% each in the Dorsal Attention Network or Frontoparietal Network (Fig. 4c and Table 1). For proportions calculated based on the mask in Fig. 3a, see Supplementary Note 3. Of note, transformation increased with greater Summary bias in the precuneus, retrosplenial cortex, bilateral lateral parietal cortex, and right prefrontal cortex. The effect was significant in the reverse direction for two parcels, located in the left medial temporal cortex and right ventral temporal cortex; i.e., for these two parcels, scenes with lower Summary bias scores exhibited stronger memory transformation.

## Discussion

Our memories of the past often seem to be temporally compressed versions of the original experiences. In this study, we investigated how the brain instantiates the compression of memories for naturalistic events. We hypothesized that reactivation effects during spoken retelling, previously observed in many high-level associative and visual areas, would be modulated

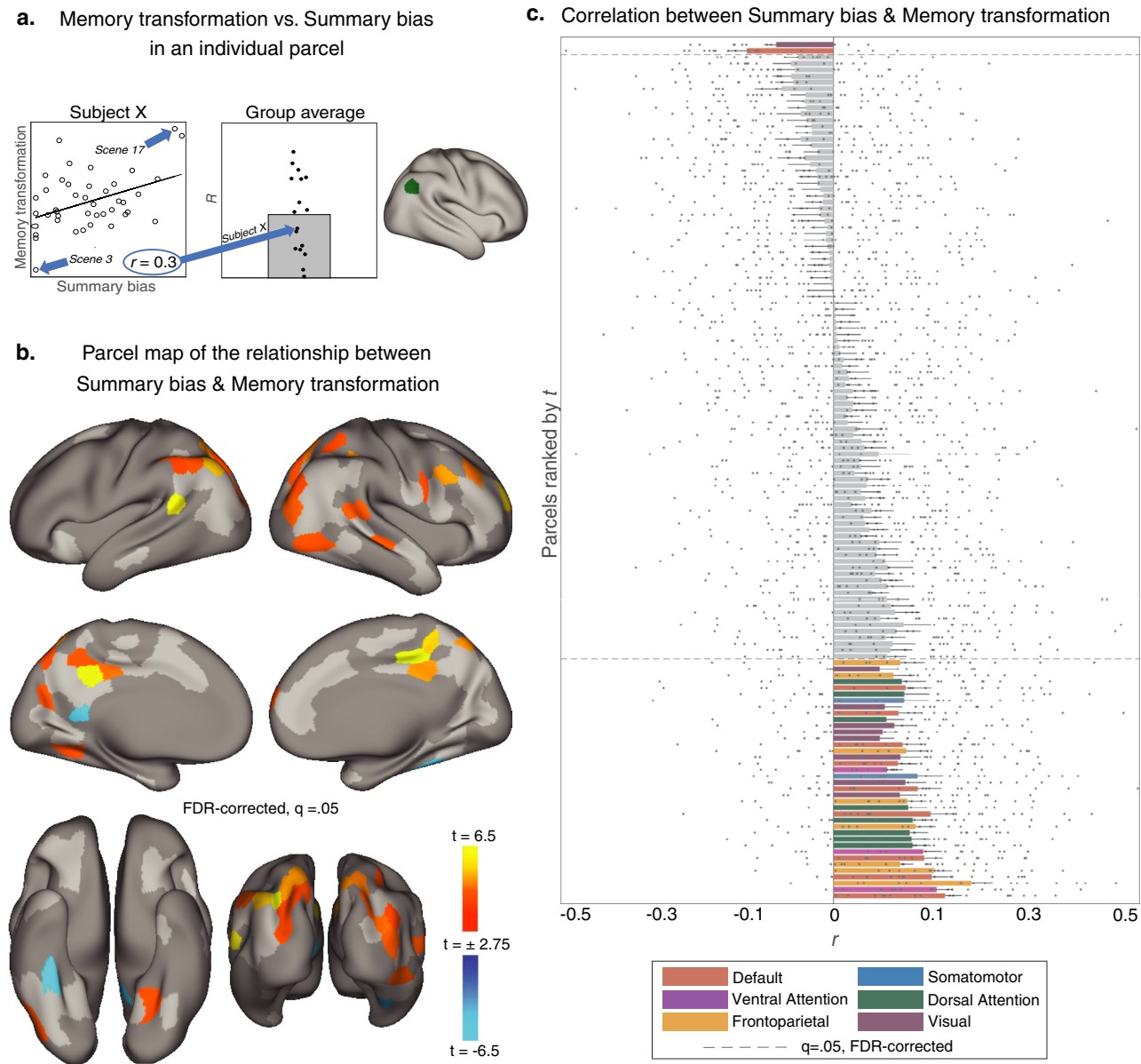

**Fig. 4 Analysis schematic and parcel-level results testing the relationship between summary bias during recall and memory transformation. a** Analysis schematic depicting how memory transformation is related to summarization in each participant and each tested brain parcel. In each parcel, the scene-level transformation values are correlated with each individual subject's Summary bias scores. The example scatterplot depicts the correlation between Summary bias and the degree of memory transformation across scenes for one subject. Subject-level correlation values are then submitted to a random-effects analysis (t-test versus zero, two-tailed). The bar plot (middle) depicts the correlation values, aggregated across subjects. The resulting t-statistic value is then assigned to the parcel on the brain map. **b** Parcel-level maps of the relationship between memory transformation and summarization. This relationship was only tested in parcels that showed reliable memory transformation effects (136/400 parcels, see Fig. 3a). Memory transformation scaled with summarization in 40/136 parcels (colored). In 96/136 of the tested parcels, this relationship was not statistically reliable (light gray). 264/400 parcels were outside the memory transformation mask and were excluded from analysis (dark gray). **c** Relationship between summarization and memory transformation across the 136 memory transformation parcels. Bar color indicates the functional network of each above-threshold parcel. Bars are sorted by the t-statistic from the random-effects analysis on the subject-level correlation values. Individual dots for each bar show subject-level correlation values and error bars indicate the standard error of the mean.

by the degree to which people temporally compress experienced events during their recall. To test this idea, we first developed a rubric to score individual utterances spoken by participants during their unguided verbal recall of a recently viewed movie. The rubric assesses the degree to which a given utterance refers to a specific moment of the movie (Temporally Precise), as opposed to describing a longer period of time (Summarizing). We found that Summary utterances during recall were associated with greater dissimilarity between encoding and recall in PMC and to a

lesser extent other regions of the DMN, relative to Temporally Precise statements. This observation agrees with prior studies showing that reinstatement (encoding-recall similarity) is positively related to subjective vividness[36] and the amount of detail of the retrieved information[44,45]. Our results extend these findings by showing that naturally varying speech content—in particular, signatures of temporal compression during a recall—also modulate reinstatement effects. However, we argue that raw dissimilarity alone is an incomplete measure, as it incorporates both

interesting changes (e.g., behaviorally-relevant shifts in representations of specific items or events) and uninteresting changes (e.g., broad differences between brain states at encoding versus at recall, and random decay). Thus, in a complementary analysis, we tested the relationship between summarization during recall and the encoding-to-recall transformation of brain activity patterns—a measure which selectively identifies changes which are consistent across subjects[2]. The results showed that, at the scene level, a bias toward summarization in recall speech significantly predicted the magnitude of transformation in a number of high-level visual and associative areas, with a plurality of parcels found in the DMN. Importantly, as the changes were shared across subjects, this result demonstrates that temporal compression during recall does not merely result in noisier reinstatement; rather, neural representations of events shift between encoding and recall in a systematic, potentially meaningful way. Together, these findings elucidate how the temporal compression of memories for naturalistic events manifests in brain activity.

The act of recollecting an event from memory typically takes less time than the original event duration[1,3,4,46]. Why should the memory of real-world events be a compressed version of the original? In addressing this question, it is important to distinguish between the compression of the *stored* information and of the *retrieved* information. In the current study, we measured brain activity during encoding and during retrieval; this allowed us to test ideas about retrieved information but did not give us access to the representational format of information stored in memory. Compression makes intuitive sense as a strategy for efficient storage—there is no need to remember everything, only what is required to meet future demands[47]. By analogy, computer algorithms for compressing image files succeed by discarding information that is not needed, while still meeting the demand that the compressed image will appear similar to the original[48]. In the case of JPEG, for example, as the human eye discriminates image brightness more finely than color, compression involves downsampling color information while brightness information is retained at a higher resolution.

On top of what has been modified in the *stored* memory representation, the demands of a given task can lead to further compression in the *retrieved* information: e.g., telling someone the steps of a recipe versus describing the atmosphere of the cooking class where you learned the recipe. Prior work has demonstrated that episodic memory retrieval can be directed or biased by the person's task demands and goals[14,49]. In the case of our task, the instructions were to retell the story (the plot) in as much detail as possible, as if telling a friend. Thus, people were mainly conveying the plot of the story and some pertinent sensory features. If subjects had been probed with questions demanding more detail, most likely they would be able to produce more detail[50]. However, the current task did not require that; instead, participants naturally varied the level of detail they provided across utterances. Our Temporal precision and Summary scores, and related behavioral measures which estimate descriptive precision during recall[51,52] indicate only the level of detail retrieved at the time of the test, not of the stored information. Future work using memory probes which more exhaustively probe stored memories could investigate the gap between what is stored and what is retrieved.

Why should the compression scheme be similar across people? We suggest three reasons. First, compression of stored or retrieved memories could be similar across people simply because brain anatomy and function are similar across people. In comparative studies, learning behavior for a given task is more variable across species than within species[53]. Second, beyond shared anatomy, participants also have shared past experiences to some degree; several studies have shown that shared past experiences

lead to shared neural responses to narrative stimuli in DMN regions. For example, Parkinson and colleagues (2018) showed that inter-subject similarity in the posterior medial cortex during movie-viewing was predicted by proximity in a social network, which the authors attribute to homophily: friends are likely to have similar prior experiences[54]. However, an extensive shared past is not needed to predict variation in inter-subject similarity: Yeshurun and colleagues (2017) showed that, when subjects were presented with an ambiguous story, a single disambiguating paragraph was presented just before the story was sufficient to push subjects into distinct groups[55]. Thus, a follow-up question for the current study is: given that people evince some similarity in how they compress memories of events, to what extent is this influenced by shared prior experiences? Finally, a third reason that memories might follow a similar compression scheme across people is that such commonality could facilitate communication between individuals. Given a shared experience by a group of individuals, later conversation between the individuals may be more coherent if they have a common ground[56,57] based on similarly remembered, forgotten, or otherwise transformed memories. This point may be particularly pertinent in our paradigm, as participants recounted their memories orally, a delivery format that may have evoked a mode of thinking associated with interpersonal communication.

What is the problem with raw dissimilarity as a measure of compression, and what is provided by the complementary transformation measure? It has been widely demonstrated that episodic remembering is accompanied by reinstatement in a set of brain regions which typically include the default mode network and high-level sensory areas; reinstatement is calculated as a similarity score between the brain activity pattern at encoding versus retrieval[32]. As "compression" implies that retrieval will be different in some way from encoding, a natural prediction would be that the more compressed a retrieved memory is, the more dissimilar its neural representation should be to that observed during encoding. Indeed, we did see this relationship in PMC (Fig. 2). However, a raw dissimilarity measurement is problematic because it will be driven up by multiple types of non-signal which cannot be discriminated from each other: broad differences between brain states at encoding vs. at retrieval, random decay of memory traces during the delay between encoding and retrieval, and other measurement or machine noise[37]. Dissimilarity will also be driven up by both subject-consistent and subject-idiosyncratic changes, both of which are interesting signals which experimenters would like to measure; but again, these cannot be separated from task-irrelevant noise. Thus, we applied a complementary test which selectively measures neural changes between encoding and recall[2] to selectively identify subject-consistent changes, separating them from state-related changes, subject-idiosyncratic changes, and task-irrelevant noise.

Chen et al. (2017)[2] reasoned that if retrieved activity patterns are more similar *between individuals* than to the originating pattern at encoding, then it can be inferred that patterns were transformed in the interval between encoding and retrieval in a systematic (non-noise) manner. In the schematic shown in Fig. 5, brain activity patterns are visualized as vectors in multidimensional space, where each dimension corresponds to a brain voxel; we propose functional labels for different types of pattern shifts.

**Noise and subject-idiosyncratic shifts**. For a given event, the retrieved event pattern may differ from the original event pattern due to (1) task-irrelevant noise, and/or (2) the idiosyncratic way in which an individual recollects the event; 1 and 2 cannot be distinguished. In this scenario (Fig. 5, Model 1), the direction in which the Recall pattern shifts away from the original Movie

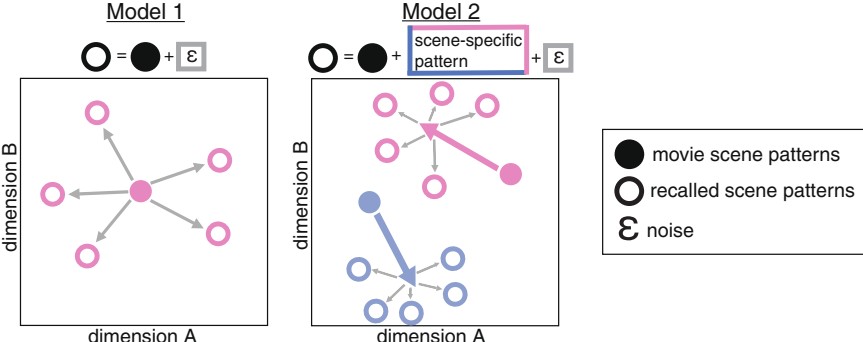

**Fig. 5 Schematic of possible contributions to the pattern changes observed between encoding and recall of a movie scene.** Each solid dot represents the neural pattern evoked by a single scene during movie viewing, here shown in a hypothetical two-dimensional space. The rings of the same color depict the neural patterns evoked when different subjects recall the scene. In both proposed models, a scene's pattern change from movie to recall is partly due to noise or subject-idiosyncratic shifts, shown as gray arrows. In Model 1, the pattern shifts from movie to recall are entirely explained by either noise or subject-idiosyncratic shifts (the two are indistinguishable). In Model 2, the pattern change is consistent across people, but varies by a movie scene, such that the magnitude and direction of the shift (colored arrows) depends on the features of each movie scene. Noise/subject-idiosyncratic shifts (gray arrows) are also present.

pattern varies across individuals. The similarity between different individuals' recalled scene patterns must be lower than the similarity between recall and movie patterns (in Fig. 5, Model 1, the average distance between rings must be greater than the average distance between the rings and the filled circle).

**Event-specific shifts**. Alternatively, the retrieved event pattern may differ from the original event pattern in a common way across people, such that each person's recall pattern shifts away from the original pattern in the same direction for a given event, but in different directions for different events (Fig. 5, Model 2). With the addition of this cross-subject-consistent shift, it becomes possible for recall patterns to be more similar to each other than they are to the original patterns (in Fig. 5 Model 2, for a given color, the average distance between rings is smaller than the average distance between the rings and the filled circle). These event-specific shifts reflect differences in the stimulus features across events that in turn affect their transformation (one component of which could be compression) between encoding and recall. For example, imagine that for Event A the most memorable features are foreground objects, while for event B the most memorable feature is the spatial layout; and that 50% of voxels are sensitive to object category, while the other 50% are sensitive to the spatial layout. In this case, the A and B event patterns would shift in different directions, but the direction of the shift would be the same across people within each event (A or B).

In the current paper, we show evidence that event-specific shifts are predicted by behaviorally-expressed summarization, or temporal compression, during recall; the degree of compression determines the degree of observed pattern transformation in a number of parcels throughout high-level visual and associative areas, with a plurality of parcels located in the DMN.

**Future directions**. Although these inter-subject comparisons enable the identification of pattern changes that are shared across subjects, they do not capture pattern changes that are specific to individual subjects, which includes both a subject-specific signal component and a subject-specific noise component. Recently, there has been growing interest in measuring such subject-idiosyncratic differences using other methods such as inter-subject representational similarity analysis (IS-RSA)[58–62]. Using IS-RSA, researchers can examine whether second-order statistics among individuals (e.g., their similarity relations along a

behavioral dimension) correspond to the geometric mapping of brain pattern similarities across individuals. For example, while watching an animation of abstract shapes[61] and listening to an audio story[63], subject pairs who expressed more similar interpretations of the stimulus also showed increased ISC in several brain regions. A promising avenue of exploration for future work could be the quantification of separate contributions for subject-shared and subject-idiosyncratic signals during naturalistic encoding and recollection. Future studies could also benefit from the inclusion of larger datasets, as the relatively small sample size in the current dataset ($n = 17$) limited our ability to estimate the test-retest reliability of the between-subjects similarity measures (see Supplementary Fig. 3).

Changes in event representations between encoding and recall, whether they are subject-idiosyncratic or subject-consistent, are likely not limited to compression. We envision compression as an operation wherein representations are selectively pruned, and perhaps guided by previously learned templates or schemas[1]. Alongside the compression of the encoded experience, new information could also be incorporated into the retrieved memory, e.g., the representation could be integrated or supplemented with related prior episodes[64–67]. Such changes would also contribute to event-specific shifts (see Fig. 5: changes could be modeled as new information being added to recall event patterns in a subject-consistent manner) but would not be captured in our Summary/Temporally Precise ratings. In the current study, we focused on compression due to what we observed in the behavior: on average, 76% of participants' utterances were judged by raters to be literal and accurate (either Summary or Temporally precise), leaving 24% of utterances in the Other category, which included inaccurate and metacognitive statements, as well as elaborations and inferences which might be considered new information. Exploration of additional operations embedded in the changes between neural event representations at encoding and recall could be pursued with paradigms in which subjects are specifically instructed, e.g., to compress, integrate, or add information during retrieval[68].

In conclusion, this work provides evidence that the temporal compression of remembered events manifests in brain activity in a distributed set of high-level visual and associative areas. During temporally compressed recall, individuals' speech content and brain patterns showed decreased similarity to the original movie events, indicating that the summarized memories are altered versions of the initial experiences. In order to address possible

problems with raw dissimilarity measures, we additionally assessed encoding-to-recall pattern changes that were consistent across subjects; even under these restricted conditions, we observed a significant relationship between temporal compression and neural pattern changes in many of the same high-level regions, with a plurality of the identified parcels falling within the DMN. These findings illuminate how our memories of real-world events diverge from the original experiences in a consistent manner across people and reveal the neural transformations that accompany the summarization or compression of our experiences.

## Methods

**Participants**. Twenty-two adults (10 female, ages 18–26) were recruited from the Princeton community. Participants were recruited using flyers posted on the Princeton University campus and an announcement posted on a university-affiliated participant recruitment website. All participants were right-handed native English speakers who reported normal or corrected-to-normal vision and had not watched any episodes of Sherlock prior to the experiment. Participants were randomly sampled from those who met the age, language, and handedness criteria. In addition, all participants had not watched any episodes of Sherlock before the experiment. No statistical methods were used to predetermine the sample size, and the sample size was similar to those reported in previous publications. Participants provided written consent prior to the experiment in accordance with the Princeton University Institutional Review Board. Data from five participants were discarded due to excessive head motion, short recall (less than 10 min), or falling asleep during scanning. Data were collected between June and December 2013, and no participants declined participation. This dataset was originally reported in Chen et al., (2017)[2].

**fMRI acquisition**. Whole-brain anatomical and functional MRI data were collected on a 3 Telsa Siemens Skyra scanner. Anatomical images were acquired with T1-weighted MPRAGE pulse sequences (0.89 mm$^3$ resolution). T2*-weighted functional images were collected with a gradient echo-planar imaging sequences in 27 ascending interleaved slices with $3 \times 3$ mm$^2$ voxels and a 1.5-s TR (echo time: 28 ms, flip angle: 64 degrees, field-of-view: $192 \times 192$ matrix, slice thickness: 4.0 mm). The Psychophysics Toolbox (http://psychtoolbox.org/) for MATLAB was used to display the movie and to synchronize stimulus onset with MRI data acquisition. Participants' speech was recorded using a customized MR-compatible recording system (FOMRI II; Optoacoustics Ltd.).

**Procedure**. Participants completed three scanning runs. During the first two scanning runs (23 and 25 min in duration, respectively), the participants viewed the first episode of the BBC television series *Sherlock*. During the third scanning run, participants were instructed to retell the events of the episode in their own words, as though they were describing the episode to a friend. During verbal recall, participant responses were recorded via microphone and transcribed. The duration of the third scanning run varied according to how long each participant spoke but lasted at least 10 min. Motion was minimized by instructing participants to remain very still while speaking and by stabilizing participants' heads with foam padding. Artifacts generated by speech may introduce some noise, but they cannot induce positive results, as the inter-subject correlation analyses depend on spatial correlations between sessions (movie–recall or recall–recall). Data were later segmented into discrete events according to plot/character changes, as determined by an independent annotator who was blind to the experimental hypotheses.

**Behavioral data processing**. Transcripts were written of the audio recording of each participant's spoken recall. Timestamps were then identified that separated each audio recording into the same 50 scenes that had been previously selected for the audiovisual stimulus. A scene was counted as recalled if the participant described any part of the scene. Scenes were counted as out of order if they were initially skipped and then described later.

**fMRI preprocessing**. Preprocessing was performed using FSL (http://fsl.fmrib.ox.ac.uk/fsl) and custom MATLAB code, and included motion correction, slice time correction, linear detrending, high-pass filtering (140 s threshold), and coregistration and affine transformation of the functional volumes to a brain template (MNI, Montreal Neurological Institute, standard). Functional images were resampled to a spatial resolution of 3 mm isotropic voxels. All calculations were performed in volume space. As a final step, projections onto a cortical surface for visualization were performed using the Human Connectome Project (HCP) workbench (https://www.humanconnectome.org/software/connectome-workbench).

**Movie event boundaries**. The movie (Sherlock Episode 1) was segmented into individual events at two different levels of granularity. For the coarse-grained

segmentation, the movie was split into 50 scenes, where scene boundaries followed major narrative shifts[2]. Coarse scenes were on average 57.7 s long (SD = 41.6 s, max = 180 s, min = 11 s). The fine-grained segmentation consisted of 1000 micro-segments, including the original 50-scene boundaries. Fine-grained micro-segments lasted 3.95 s on average (SD = 2.20 s, max = 19 s, min = 1 s).

**Utterance coding by direct judgments**. Each participant's speech from the recall session was transcribed. The transcribed speech was then divided into individual utterances, where each utterance roughly corresponded to a single complete sentence. For long sentences that contained three or more subordinate clauses, sentences were broken up into multiple utterances. Utterance boundaries were subjectively defined as breaks in the participant's speech. These breaks could occur, for example, during pauses in speech, or during shifts in the recalled content, such as when the speaker switches to a new topic. Each utterance was comprised of 14 words on average (SD = 2.4), and the mean utterance duration was 6.83 s (SD = 1.6 s). The total duration of the recall session varied across participants, as did the total number of recall utterances, ranging from 114 to 468 utterances across participants (average = 203.1, SD = 110.1). For additional descriptive statistics see Supplementary Table 1.

Each utterance was coded according to (1) the movie events it described and (2) the amount of time that it took for the described events to elapse in the movie (i.e., the degree of temporal precision). First, for the movie scene coding, each utterance was assigned a Start movie segment and an End movie segment label. The scene labels were assigned according to the fine-grained, 1000 micro-segmentation. The labeling of each utterances starting and ending movie scenes was completed by a single annotator with no knowledge of the experimental design or hypotheses[2].

Next, for temporal precision, each utterance was labeled as either Summary, Temporally Precise, or Other (Fig. 1). These labels were generated in two distinct ways: Direct Judgment and Automatic. The two methods yield consistent labels across subjects, r = 0.60 (min: r = 0.39, max: r = 0.77, SD = 0.12). In addition, the automated labeling method can provide a continuous measure of temporal compression, as opposed to the binary summary/temporally precise distinction. The main reported results use the Direct Judgment labels; we describe the procedure for generating these labels below.

For the Direct Judgment labeling, participants' speech during recall was categorized into the three non-overlapping categories (i.e., Summary, Temporally Precise, and Other). Literal and accurate statements about the movie content were categorized as Temporally Precise if they describe movie events that elapsed in less than ten seconds. In contrast, literal and accurate statements were classified as Summary if they described events that took longer than ten seconds to occur. Only literal and accurate utterances were coded; utterances that contained factually incorrect information (i.e., confabulation) or that referred to events other than the movie content (i.e., not recall) were coded as Other and excluded from subsequent analysis.

The temporal precision labeling was completed by two coders (including author EM) who were provided the definitions for Temporally Precise and Summary utterances listed above. Each coder worked separately to label all utterances according to these definitions. After completing the labels for each participant, the two coders met and identified all utterances for which their labels did not match. The raters originally had a 77% agreement rate. For utterances with mismatching labels, the coders then reviewed the utterance content and re-watched the corresponding movie scenes together to arrive at a consensus label.

All participants produced utterances in all three categories (Fig. 1c). On average, 47% of participants' utterances were coded as Temporally Precise and 30% as Summary. Across participants, an average of 24% of utterances were labeled as Other and excluded from subsequent analyses (Fig. 1).

To compare the relative amounts of the different speech categories present in each subject's recall, we counted the number of words included in each category, summing across all of a subject's statements (Fig. 1b). Most participants produced a greater proportion of Temporally Precise versus Summary words (11/17 participants), although the number of words and the relative proportion of utterances in the different categories varied across individuals.

In contrast to the Direct Judgment method, the Automatic method considers the micro-segment(s) assigned to each recall utterance and then sums the duration of these segments to determine whether each utterance's described movie events elapsed in less than or greater than ten seconds. Additional details for the Automatic method, and details for measuring temporal compression as a continuous measure (termed temporal compression factor, or TCF), are described below.

**Utterance coding by automatic methods**. The behavioral and fMRI results described in the Results section were based on Direct Judgments of temporal precision, in which human raters labeled recall utterances based on whether the described events elapsed in more than ten seconds (Summary) or less than ten seconds (Temporally Precise). These labels can also be assigned with Automatic methods, as described below.

*Indirect/automatic*. In a complementary analysis to the Direct Judgments, the categories of Summary and Precise were determined automatically, based on the movie segment labels assigned to each statement, as judged by a coder who worked

with the data prior to the current analysis (see Chen et al., 2017 and Vodrahalli et al., 2018 for details). This method differs from the Direct Judgment approach in that the Direct Judgment coders specifically considered whether each statement referred to a 10 s window or longer, while the Automatic coder began by first segmenting the movie into 1000 events, then had the goal of assigning movie segments to each recall statement, without considering the unit of 10 s. In the Automatic method, if the total summed duration of the labeled movie segments exceeded 10 s, the statement was labeled as Summary, otherwise, it was labeled as Precise. Utterances for which no corresponding movie segments could be identified were labeled as Other.

Scene-level values of recall behavior can also be computed for each subject, akin to the Summary bias scores reported using the Direct Judgment method in the main manuscript. To compute these values, we repeated the steps described for Direct Judgment as follows: for each of the 50 movie scenes, we identified the recall utterances that described the scene (after translating the utterance-level movie event labels from the 1000 micro-segment level to the 50-scene level). Then, we counted the number of words (excluding stop words) contained in these utterances and calculated the proportion of words that came from utterances that had been automatically labeled as Summary. The list of stop words excluded from the word counts are provided in Supplementary Note 2.

*Temporal compression factor.* The Indirect/Automatic approach also enabled us to calculate a temporal compression factor (TCF) for each recall utterance. For each utterance, we computed the number of spoken words (excluding stop words) and then divided this value by the total duration of the described movie events, which is the same duration described in the Automatic method above. Thus, the TCF value is a ratio, where the numerator quantifies the amount of information the participant devoted to describing the events, and the denominator quantifies how long it looks for these events to elapse in the movie. This value is subtracted from 1, so that higher TCF scores indicate increased temporal compression, or that fewer words were spent relative to the amount of time passed.

To compute scene-level TCF values for each subject, we identified the utterances that described each scene (scaling from the 1000 micro-segment level up to the 50-scene level), and then calculated the scene-average TCF value by averaging across the utterance-level TCF values.

Thus, the three different measures (Direct Judgment, Automatic, and Temporal Compression Factor) can each provide scene-level values of recall behavior for each subject (e.g., Summary bias scores for each scene, as described in the main manuscript). The three measures were positively correlated in each subject, as should be expected given that are variations of the same judgment (i.e., degree of temporal compression at recall relative to encoding). The Direct Judgment ratings of Summary bias are moderately correlated with both the Automatic ratings ($r = 0.55$, SD = 0.18, min = 0.19, max = 0.84) and the TCF scores ($r = 0.41$, SD = 0.15, min = 0.11, max = 0.64), while the TCF and Automatic ratings are correlated at $r = 0.68$ (SD = 0.09, min = 0.51, max = 0.85). A parcel-level map of the relationship between TCF and memory transformation is shown in Supplementary Fig. 4.

**Text-based analysis of the similarity between movie annotations and summary and precise recall.** In order to compute the overlap between movie and recall content, and to test how movie-to-recall text similarity varied by recall type (i.e., for Summary versus Temporally Precise recall utterances), we performed an utterance-level analysis comparing the speech generated at recall against the original movie annotations. All movie segment annotations and recall statements were converted into unique 512-element vectors using the Universal Sentence Encoder algorithm[42]. We computed the similarity between each recall statement vector and its corresponding movie segment vector(s). If a recall utterance described multiple segments, a separate cosine similarity score was computed for each of these segments and then averaged together to yield one composite cosine similarity score for the utterance. Average movie versus recall vector similarity was then computed separately for each participant and each recall type (i.e., movie segments vs. Summary utterances and movie segments vs. Precise utterances).

**Within-subjects reinstatement for summarized versus temporally precise recall.** Memory reactivation was measured at the individual-subjects level by computing pattern similarity between each micro-segment during movie viewing and subsequent recall utterance(s) that described each micro-segment. If several recall utterances described the same micro-segment, the patterns for these utterances were first averaged together prior to computing the reinstatement value. Reactivation scores were calculated separately for segments that were later recalled in a Temporally Precise manner versus Summarized. If a micro-segment was later described with both Summary and Temporally Precise utterances, it received two separate reactivation scores.

For the ROI-based analysis, pattern similarity was computed in voxels located in a bilateral PMC ROI (see Fig. 3a and Supplementary Table 2 for ROI definition). In a parcel-based analysis, reinstatement was computed separately for Summary and Temporally Precise recall in every parcel in the brain. To create a contrast map of Summary versus Temporally Precise reinstatement at the parcel level, we limited our analyses to parcels that showed a reliable Summary or Temporally Precise reinstatement effect (283/400 parcels, see Fig. 2d). No parcels survived FDR

correction at $q = 0.05$; at a more liberal threshold ($q = 0.10$), ten parcels showed marginal effects of greater reinstatement for Temporally Precise versus Summarized recall.

**Computing degree of summarization by a movie scene.** We computed the average degree of temporal precision for each movie scene in each subject. First, for each participant, we sorted the utterances according to the movie scene(s) that they referred to. Note that the same utterance could be assigned to several different movie scenes, depending on its Start movie scene and Stop movie scene labels. Then, we counted the total number of words spoken about each movie scene by summing across all words in all of its corresponding utterances. We then sorted each scene's utterances by temporal precision, and removed "stop" words (e.g., "a", "is", "the") and non-words (e.g., "uh", "um") from all utterance transcriptions (see Supplementary Note 2). Finally, we counted the proportion of words that were assigned to each label (i.e., Summary, Temporally Precise, or Other). For each participant, this analysis yielded a one %Summary bias score for each movie scene that the participant recalled. Movie scenes that were recalled by less than five out of the 17 participants were excluded from this analysis.

**Memory transformation: methods (parcels).** We tested for brain areas that contain scene-specific content that is shared across individuals in each of 400 parcels from an independent whole-brain resting-state parcellation[43]. In each parcel, we first performed the scene-level, between-subject analyses, computing recall-to-recall similarity and movie-to-recall similarity for each scene[2]. The movie-to-recall analysis uses the fMRI data acquired during both the movie-viewing session and the subsequent free recall session. Each timepoint (TR) was labeled according to the scene that the subject was currently viewing (during encoding) or describing (during recall). The constituent TRs in each scene were then averaged together, yielding 50 scene patterns for the movie data in each subject. For the recall data, the number of scene patterns varied across subjects, depending on which scenes each subject mentioned during recall. For the between-subjects movie-to-recall analysis, each individual's movie scene patterns were compared to the corresponding recall patterns for that same scene, averaged across all of the other subjects who recalled that scene. This analysis yields two values for each parcel: one indexing the scene-average movie-to-recall similarity in the parcel, and one for the scene-average recall-to-recall similarity. We then constrained our analyses to parcels that showed significant effects in either the movie-to-recall or recall-to-recall comparisons. Of the 400 parcels, 174 met this criterion (all light gray and hot colored parcels in Fig. 3). For these statistically reliable parcels, we identified the ones where between-subjects recall-to-recall similarity exceeded movie-to-recall similarity. The difference between these values quantifies the memory transformation value for that parcel.

**Linking transformation and summarization: methods.** To test for relationships between the degree of summarization and memory transformation, we focused on the parcels that showed memory transformation effects. In these parcels, we correlated each subject's scene-level %Summary bias scores with the group-average memory transformation value for each scene.

**Statistics and reproducibility.** For the text-based analysis comparing the similarity between USE vectors for the movie annotations and spoken recall, significance was evaluated using paired t-tests (two-tailed) to compare cosine similarity for vectors of matching segments (i.e., movie segments and recall utterances that covered the same content) versus mismatching segments across all participants ($n = 17$). This was performed separately for each recall type (i.e., movie segments versus temporally precise recall utterances, and movie segments versus summarized recall utterances). To directly compare participants' average movie-to-recall cosine similarity for Summary versus Temporally Precise recall segments, the cosine similarity values for the matching segments of each recall type were submitted to a paired t-test (two-tailed). These t-tests were deemed significant at $p < 0.05$ (two-tailed).

For the within-subjects reinstatement analysis in the PMC ROI, the resulting reinstatement values across subjects ($n = 17$) were submitted to a one-sample t-test versus zero (two-tailed), computed separately for Summary and Temporally Precise Reactivation. To directly compare reinstatement across recall types, each subject's two reactivation values were submitted to a paired t-test and deemed significant at $p < 0.05$ (two-tailed). For the parcel-based analysis, one-tailed $p$ values were computed at each of the 400 parcels by comparing the group-average reinstatement value to a null distribution that was generated by randomly shuffling the segment-to-utterance correlation matrices 1000 times. To determine thresholding and correct for multiple comparisons, the parcel-level $p$ values were submitted to false detection rate (FDR) correction ($q = 0.05$). Statistical significance and FDR correction were computed separately for the Summary reinstatement and Temporally Precise reinstatement maps.

To compute the parcel map that contrasts Summary versus Temporally Precise within-subject reinstatement, analyses were limited to parcels that showed a reliable Summary or Temporally Precise reinstatement effect (283/400 parcels, see Fig. 2d). Subsequent FDR correction was limited to parcels included in this mask. At each parcel in the mask, we directly compared the two reinstatement values

across subjects (*n* = 17) with a paired *t*-test (two-tailed). No parcels survived FDR correction at *q* = 0.05; at a more liberal threshold (*q* = 0.10), ten parcels showed marginal effects of greater reinstatement for Temporally Precise versus Summarized recall.

For the inter-subject pattern similarity analyses (i.e., movie-to-recall and recall-to-recall), statistical significance was determined by shuffling the scene labels 1000 times to generate a null distribution. For each analysis, a one-tailed *p* value was calculated as the proportion of values from the null distribution that were equal to or greater than the observed similarity values for matching scenes. The resulting *p* values were then corrected for multiple comparisons using FDR correction (*q* = 0.05). To test for reliable correlation between %Summary bias and memory transformation, we submitted each participant's (*n* = 17) scene-level correlation between %Summary bias and memory transformation values to a one-sample *t*-test versus zero (two-tailed). This test was performed at each parcel included in the memory transformation mask, which was limited to parcels with either significant movie-to-recall or significant recall-to-recall similarity. Across parcels in the memory transformation mask, FDR correction (*q* = 0.05) was then applied to determine statistical thresholding.

**Citation diversity statement**. Recent work in several fields of science has identified a bias in citation practices such that papers from women and other minority scholars are under-cited relative to the number of such papers in the field[69–77]. Here we sought to proactively consider choosing references that reflect the diversity of the field in thought, form of contribution, gender, race, ethnicity, and other factors. First, we obtained the predicted gender of the first and last author of each reference by using databases that store the probability of a first name being carried by a woman[73,78]. By this measure (and excluding self-citations to the first and last authors of our current paper), our references contain 16.2% woman(first)/woman(last), 12.92% man/woman, 24.42% woman/man, and 46.45% man/man. This method is limited in that (a) names, pronouns, and social media profiles used to construct the databases may not, in every case, be indicative of gender identity and (b) it cannot account for intersex, non-binary, or transgender people. Second, we obtained the predicted racial/ethnic category of the first and last author of each reference by databases that store the probability of a first and last name being carried by an author of color[79,80]. By this measure (and excluding self-citations), our references contain 8.14% author of color (first)/author of color(last), 15.43% white author/author of color, 18.88% author of color/white author, and 57.56% white author/white author. This method is limited in that (a) names, Census entries, and Wikipedia profiles used to make the predictions may not be indicative of racial/ethnic identity, and (b) it cannot account for Indigenous and mixed-race authors or those who may face differential biases due to the ambiguous racialization or ethnicization of their names. We look forward to future work that could help us to better understand how to support equitable practices in science.

**Reporting Summary**. Further information on research design is available in the Nature Research Reporting Summary linked to this article.

## Data availability

In addition, the data and materials that support the findings of this study are available on the Open Science Framework data and code repository titled "Neural Signatures of Compression in the Retelling of Past Events" at the following URL: https://osf.io/8gmes/ The preprocessed fMRI data are available for download at: https://dataspace.princeton.edu/handle/88435/dsp01nz8062179.

## Code availability

The custom MATLAB computer code for reproducing the parcel maps and all depicted figures in this study are available for download on the Open Science Framework data and code repository titled "Neural Signatures of Compression in the Retelling of Past Events" at the following URL: https://osf.io/8gmes/. The code and the corresponding data[81] are also available for download at the following URL: https://doi.org/10.5281/zenodo.6382857.

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

## Acknowledgements
We thank James Antony, Buddhika Bellana, and Christopher Honey for their comments on an earlier version of the manuscript. We thank Qingwei Zhang for assistance with behavioral ratings and Yoonjung Lee for her feedback on the data and analysis materials available on the Open Science Framework. This work was supported by a National Science Foundation Postdoctoral Research Fellowship (SBE SPRF 1911650 to E.M.) and a Sloan Research Fellowship (to J.C.).

## Author contributions
E.M. and J.C. conceived and designed the experiments. E.M. and J.C. performed the experiments and analyzed the data. E.M. and J.C. wrote and edited the paper.

## Competing interests
The authors declare no competing interests.
