## [Peer Review File · Communications Biology]

Reviewers' comments:

Reviewer #1 (Remarks to the Author):

Thank you for inviting me to review this manuscript by Musz and Chen, in which the authors conduct a sophisticated analysis of a unique dataset that invited participants to first watch a movie, and then subsequently recall the key narrative features of the movie (while also in the MRI scanner). The authors hypothesized that the 'compression' of the BOLD patterns observed during the recall phases of the task predicted specific patterns of encoding-to-recall transformations in multiple regions of association cortex.

I read the paper with interest, and was regularly impressed with the apparent care and focus demonstrated by the authors. The question and hypotheses were interesting, the approaches used were sophisticated and the discussion was balanced. I have no comments that I feel would add to the manuscript, and such would simply like to commend the authors for an excellent piece of science.

Reviewer #2 (Remarks to the Author):

1. Major claim: Neural representations are not simply reactivated, but can also be transformed due to temporal compression during a universal form of human memory expression: verbal retelling.
2. Novelty: The first observation that temporal compression in individuals' retelling of past events predicts systematic encoding-to-recall transformations in a number of higher associative regions.
3. Field influence: The scheme of movie-watching and its new development of analytical methods could be a promising impact on the field of task-based fMRI, which recently has been concerned with limited measurement reliability of measuring the individual differences.
4. Reliability: Is there any previous work on the test-retest reliability evaluation of the proposed fMRI measurement? If no, it would be a nice add to do such reliability assessment. This is because the reliability is an important factor for experimental design (e.g., see 10.1038/s41562-019-0655-x for a commentary).
5. Validity: It seems the authors focused their analyses on cortical surfaces. Regarding the important role of hippocampus in functioning human memory, what is the role of the hippocampus in the context of the current work? A recent work has reported anterior HIPP and posterior HIPP connects to different cortical networks (PMN versus DMN) (doi: 10.1073/pnas.2101743118). This might be an interesting point of showing predictive validity during revision.
6. Statistics: The sample size is somehow limited, especially for individual differences. How the authors determined the sample size? Please report all the statistics such as power, effect size, confidence interval and others for specific types of tests.

Reviewer #3 (Remarks to the Author):

The authors recorded brain activity when participants were retelling the movie scenes they've watched in the scanner. They found during temporally compressed recall, individuals' speech content and brain patterns showed decreased similarity to the original movie events. They also found temporal compression in individuals' retelling of past events predicts encoding-to-recall transformations. These findings are interesting for understanding neural representations during a universal form of human memory expression: verbal retelling. I have some concerns regarding to the current manuscript as follows.

1. After reading the manuscript carefully, I am still confusing how did the authors calculate "pattern similarity" (Line 706). I guess this is a spatial Pearson's correlation across all the voxels within the PMC ROI (or within a parcel). But what are the values for the "pattern" of the voxels for this ROI? An utterance may have a duration on average of 6.83s, so there might be 5 time points for a given utterance. Did the authors simply average the 5 time points to get the "pattern"? I.e., the value of each voxel was a mean value of the 5 BOLD values across the time points?
2. Another way is to define each utterance as a regressor, and then perform a task activation GLM analysis. Then the beta values can be used to define the "pattern". Could the authors compare

their results with this method?

3. The authors concluded "with a plurality of the identified parcels falling within the DMN". However, for the 7 networks in Schaefer's parcellation, DMN has more parcels than some other networks. This should be taken into account. Probably should calculate the ratio of the number of identified parcels belong to DMN / the total number of DMN parcels.
4. The threshold for compression is 10s events. Please give more justification on why 10s was determined? How much would the results change if another temporal threshold was used?
5. Please provide more details of the MRI data acquisition. E.g., what's the TR?
6. Please also provide more details of the data preprocessing for the brain imaging.
7. Table 1 "Results were thresholded at the $p < .05$ level, except for rows 3-4 ($p < .10$)." Here "p" should be changed to "q" (FDR)?
8. Line 721 "(283/400 parcels, see Figure 3D)". There was no Figure 3D in the manuscript.
9. Please check the format of the references, some journals use abbreviations and some not. E.g., "Journal of Neuroscience (30)" and J. Neurosci (22).

Reviewer #1 (Remarks to the Author):

Thank you for inviting me to review this manuscript by Musz and Chen, in which the authors conduct a sophisticated analysis of a unique dataset that invited participants to first watch a movie, and then subsequently recall the key narrative features of the movie (while also in the MRI scanner). The authors hypothesized that the 'compression' of the BOLD patterns observed during the recall phases of the task predicted specific patterns of encoding-to-recall transformations in multiple regions of association cortex.

I read the paper with interest, and was regularly impressed with the apparent care and focus demonstrated by the authors. The question and hypotheses were interesting, the approaches used were sophisticated and the discussion was balanced. I have no comments that I feel would add to the manuscript, and such would simply like to commend the authors for an excellent piece of science.

We thank the reviewer for their careful reading of our manuscript and we are grateful for their thoughtful and encouraging comments.

Reviewer #2 (Remarks to the Author):

1. Major claim: Neural representations are not simply reactivated, but can also be transformed due to temporal compression during a universal form of human memory expression: verbal retelling.

2. Novelty: The first observation that temporal compression in individuals' retelling of past events predicts systematic encoding-to-recall transformations in a number of higher associative regions.

3. Field influence: The scheme of movie-watching and its new development of analytical methods could be a promising impact on the field of task-based fMRI, which recently has been concerned with limited measurement reliability of measuring the individual differences.

We are grateful for the reviewer's positive assessment of the novelty and potential field influence of our paper. They bring up a number of thoughtful and constructive comments (blue font), each of which we respond to below (black font). We have also included excerpts from the manuscript that we have added or revised, which appear in red font both in the manuscript document and in our responses.

4. Reliability: Is there any previous work on the test-retest reliability evaluation of the proposed fMRI measurement? If no, it would be a nice add to do such reliability assessment. This is because the reliability is an important factor for experimental design (e.g., see 10.1038/s41562-019-0655-x for a commentary).

We thank the reviewer for providing the reference to this interesting and timely commentary paper, and we agree that is important to determine whether our neural measure is robust and reliable. To address this, we have developed a measure to quantify the test-retest reliability. We

have added the following paragraph and figure to the Supplementary Materials as Supplementary Figure 4:

Supplementary Figure 4. There is no previous work to evaluate the test-retest reliability of our fMRI measurement of *memory transformation*, i.e., where *recall-recall* correlations (similarity of brain activity patterns across subjects during *recall*) exceed *movie-recall* correlations (similarity of *movie* brain activity patterns with *recall* brain activity patterns across subjects). To compute test-retest reliability, we split the data in half (two sub-groups with random assignment, n=9 and n=8) and computed 1) *movie-recall* similarity 2) *recall-recall* similarity and 3) the difference between 1 and 2 (i.e., *memory transformation*) in each parcel. For more details on how these measures are computed, see pg. 26. We excluded parcels that did not show reliable effects for either 1) or 2) in the whole-group data, limiting analyses to the colored and light gray parcels depicted in Figure 3a (174/400 parcels). Thus, for each of the two sub-groups, we computed the three measures in each of the 174 parcels. We then computed the Pearson correlation (r-value) between the 174 memory transformation values across the two sub-groups, which quantifies the extent to which parcel-level memory transformation values are similar across the two sub-groups. We repeated this analysis 100 times with random splits of the 17 subjects into two sub-groups. The histogram below (top row) shows the distribution of r-values (correlation of 174 memory transformation values across the two random sub-groups). The distribution average was

$r=0.14$. Reliability (the distribution average) for recall-to-recall (measure #1, middle row) was $r=.21$ and reliability for movie-to-recall was $r=.44$ (measure #2, bottom row). In sum, for all three measures, we found consistently positive correlations between the parcel values for random split-halves of the data. Note that in order to avoid performing inter-subject pattern similarity analyses with too little data, scenes were excluded if they were recalled by less than five participants in each split-half group, leading to the inclusion of an average of 13.5/50 scenes across each the 100 random samples.

This figure is now cited on pg. 13 of the Results: “For estimates of test-retest reliability for recall-to-recall similarity, movie-to-recall similarity, and memory transformation, see Supplementary Figure 4.”

While these values can provide some sense of the test-retest reliability of our neural measures, it is important to note that the split-half analysis substantially reduces the amount of data for calculation of our measures, both in terms of the sample size and the number of included data points that contribute to each measure. In our main analysis reported in the paper, using the full dataset of $n=17$, we exclude movie and recall data from any of the 50 scenes that were not recalled by at least 5/17 of the subjects, which led to the exclusion of 5/50 scenes (see grey violin plots in Figure 1B, and the dotted grey line at $x=45/50$ on the histogram below). The threshold of 5 subjects was employed to avoid performing inter-subject pattern similarity analyses with too little data, and the specific value of 5 was based on the precedent set in Chen et al. (2017), where these data were first reported. We also applied this criterion in the split-half reliability analysis, such that in computing the reliability of our three measurements, we excluded data from any scene that was not recalled by at least 5 subjects in each of the two sub-groups (i.e., 5/8 subjects for one group and 5/9 for the other group). Due to the necessarily reduced sample sizes in the split-half analysis, fewer scenes met the criterion. On average, 15.3/50 scenes were included in each analysis (see histogram below for distribution of the number of retained scenes across the 100 samples). Thus, our split-half analysis is likely more noisy relative to the original, even beyond the data reduction expected due to halving the number of subjects. In spite of this, we found consistently positive correlations between split-halves in all three measures (movie-recall, recall-recall, and memory transformation) with respect to their values across the 174 parcels.

5. Validity: It seems the authors focused their analyses on cortical surfaces. Regarding the important role of hippocampus in functioning human memory, what is the role of the hippocampus in the context of the current work? A recent work has reported anterior HIPP and posterior HIPP connects to different cortical networks (PMN versus DMN) (doi: 10.1073/pnas.2101743118). This might be an interesting point of showing predictive validity during revision.

We agree that it is worthwhile to consider how hippocampal activity might contribute to temporal compression during naturalistic recall, and how hippocampus might be involved in transforming information from perception to memory.

In an earlier paper that used the same data reported in our present manuscript, Chen et al (2017) examined the relationship between hippocampal activity during movie viewing and later recall success (“subsequent memory” analysis). In an anatomically-defined whole hippocampus ROI, for each movie scene they calculated the correlation between a given subject’s hippocampal timecourse and the average hippocampal timecourse from all other participants (temporal inter-subject correlations). They split the data by scenes that each participant later remembered versus later forgot, and found that ISC was significantly greater for remembered versus forgotten scenes. Next, they split the hippocampus into three anatomically segregated ROIs (anterior, middle, and posterior sections) and found the same effect to be significant in the anterior hippocampal ROI, but not the posterior hippocampal ROI. This finding suggests that activity during encoding in anterior hippocampus in particular contributes to successful memory.

As in the earlier paper, in the current paper we did not constrain our hypotheses to hippocampal ROIs, but rather tested effects throughout the whole brain, including parcels located in the hippocampus. Results in hippocampal parcels did not survive whole-brain statistical thresholding. In order to address the reviewer’s question, we now perform our tests in anatomically-defined hippocampal ROIs. Here, we used the same ROIs utilized in Chen et al. (2017) in the subsequent memory analysis described above.

We have added the following paragraph to the Supplementary Materials as Supplementary Note 1:

In addition to the parcel-level results reported in the main paper, we also tested for effects of 1) within-subject memory reactivation and 2) between-subject memory transformation in two anatomically-defined hippocampal ROIs: posterior hippocampus and anterior hippocampus.

For the within-subjects reactivation analysis at the micro-segment level, neither hippocampal ROI showed reliable reactivation effects during either summarized or temporally precise recall. In the posterior hippocampus, there was no reliable reactivation for micro-segments that are later recalled in a temporally precise manner, $t(16)= 1.5, p= 0.15$, for micro-segments that are later summarized, $t(16)= 1.10, p= 0.28$. This was also the case for anterior hippocampus: there was no reliable reactivation for micro-segments with temporally precise recall, $t(16)= 1.6, p= .13$, or for summarized micro-segments, $t(16)= 0.48, p= .42$.

We also tested for memory transformation effects in each hippocampal ROI, by computing encoding-to-recall similarity and recall-to-recall similarity across people at the 50-scene level, as described on pg. 25-26. In posterior hippocampus, neither the recall-to-recall (RR) nor movie-to-recall (MR) effect was significant (RR: $t(16)= -0.7, p= .52$, mean similarity:

$r = -0.004$; MR: $t(16) = 0.66$, $p = .52$, mean similarity: $r = 0.01$). This was also the case in anterior hippocampus (RR: $t(16) = 0.67$, $p = .51$, mean similarity: $r = 0.007$; MR: $t(16) = 0.36$, $p = .73$, mean similarity: $r = 0.004$). In addition, we found that recall-to-recall (RR) similarity did not exceed movie-to-recall similarity (MR) in either ROI (anterior hippocampus: RR minus MR similarity: $r = -0.05$; posterior hippocampus: RR minus MR similarity: $r = -0.01$).

There are several possible reasons why we might have failed to find effects in the hippocampal ROIs; it is challenging to speculate on the factors that contribute to a null result. However, we agree that some readers may appreciate the inclusion of these results. We now mention this note on pg. 13 of the results section: “Neither anterior hippocampus nor posterior hippocampus showed memory transformation or reliable within-subject reinstatement effects (Supplementary Note 1).”

6. Statistics: The sample size is somehow limited, especially for individual differences. How the authors determined the sample size? Please report all the statistics such as power, effect size, confidence interval and others for specific types of tests.

The sample size was predetermined, as we re-used data that was previously collected for an earlier published study (Chen et al., 2017).

For the text-based analysis (pg. 7-8), and reactivation analysis in the posterior medial cortex ROI (pg. 9), we now report effect sizes and confidence intervals in addition to the t-statistics, standard deviations, and mean values.

Page 8 now reports: “Movie vs. recall text cosine similarity was reliably greater for matching versus mismatching movie micro-segments and recall utterances; this was true for both Summary utterances, $t(16) = 32.54$, $p < 0.001$, $d = 7.89$, 95% CI [0.10, 0.11] (M cosine difference = 0.10, SD = 0.01) and Temporally Precise utterances, $t(16) = 44.77$, $p < 0.001$, $d = 10.86$, 95% CI [0.18, 0.19] (M cosine difference = 0.18, SD = 0.02). Critically, movie vs. recall text cosine similarity was significantly lower for micro-segments that were later summarized, relative to micro-segments that were precisely recalled, $t(16) = 8.35$, $p < 0.001$, $d = 2.49$, 95% CI [0.06, 0.09] (M cosine difference = 0.07, SD = 0.04); this effect was in the same direction for all individual subjects (17 of 17) (Supplementary Figure 1).”

Page 9 now reports: “Reinstatement was significant in the PMC ROI both for movie micro-segments that were recalled in a Temporally Precise manner, $t(16) = 9.49$, $p < .001$, $d = 2.30$, 95% CI [0.08, 0.12] (mean $r = .10$, SD = .04) and those that were later Summarized, $t(16) = 7.07$, $p < .001$, $d = 1.71$, 95% CI [0.04, 0.07] (mean $r = .05$, SD = .03). Furthermore, in 16/17 subjects, reinstatement was greater for Temporally Precise than for Summarized recall; $t(16) = 5.11$, $p < 0.001$, $d = 1.24$, 95% CI [0.03, 0.07], all two-tailed tests (Figure 2A).”

Reviewer #3 (Remarks to the Author):

The authors recorded brain activity when participants were retelling the movie scenes they've watched in the scanner. They found during temporally compressed recall, individuals' speech content and brain patterns showed decreased similarity to the original movie events. They also

found temporal compression in individuals' retelling of past events predicts encoding-to-recall transformations. These findings are interesting for understanding neural representations during a universal form of human memory expression: verbal retelling. I have some concerns regarding to the current manuscript as follows.

We thank the reviewer for their insightful comments and for the close attention they have given to our manuscript. Our responses (in black font) to their comments (blue font) appear below. We have also included excerpts from the manuscript that we have added or revised, which appear in red font both in the manuscript document and in our responses below.

1. After reading the manuscript carefully, I am still confusing how did the authors calculate “pattern similarity” (Line 706). I guess this is a spatial Pearson's correlation across all the voxels within the PMC ROI (or within a parcel). But what are the values for the “pattern” of the voxels for this ROI? An utterance may have a duration on average of 6.83s, so there might be 5 time points for a given utterance. Did the authors simply average the 5 time points to get the “pattern”? I.e., the value of each voxel was a mean value of the 5 BOLD values across the time points?

Thank you for highlighting the need for more clarity in the description of this analysis. Yes, the pattern is comprised of the voxel-level activity values, which for a given event, are averaged across corresponding timepoints. This is described on pg. 26: “Each timepoint (TR) was labeled according to the scene that the subject was currently viewing (during encoding) or describing (during recall). The constituent TRs in each scene were then averaged together, yielding 50 scene patterns for the movie data in each subject. For the recall data, the number of scene patterns varied across subjects, depending on which scenes each subject mentioned during recall.” This approach is frequently used to compute event-level pattern similarity while participants watch and recall naturalistic stimuli (e.g., Chen et al., 2017; Zadbood et al., 2017; Baldassano et al., 2017).

In order to clarify this point for readers, we added the following to pg. 12 of the Results: “We first identified and averaged across the timepoints (TRs) that corresponded to each of the 50 movie scenes for movie-viewing and for recall, and then computed the parcel maps for movie-to-recall (MR) similarity and recall-to-recall (RR) similarity across people.”

2. Another way is to define each utterance as a regressor, and then perform a task activation GLM analysis. Then the beta values can be used to define the “pattern”. Could the authors compare their results with this method?

We thank the reviewer for this suggestion, and we understand the desire to compare our current methodology to traditional analytic approaches in the literature. However, we believe that our data are not suited for GLM-style analysis. For fMRI data collected during continuous dynamic stimuli (movies) or speech, analysis is not typically performed with a GLM because stimulus presentation is not punctate or discrete; that is, there is no reason to assume that each “utterance” is an impulse that will elicit an HRF-like response in the brain. Inter-subject correlation methods were developed specifically to surmount this problem: “...by using one individual's neural

activity pattern as the ‘model’ for the neural activity of other individuals during a common stimulus or psychological state, we aim to circumvent the need to generate a model from the stimulus itself” (Hasson et al., 2004; Hasson et al., 2010). (In this context, “generating a model from the stimulus” corresponds to the GLM / design matrix approach used for more traditional experiment designs.) More recently, the inter-subject correlation method has been expanded to signals in the temporal domain to those in the spatial (multi-voxel pattern) domain, with the same underlying logic: “By leveraging one individual’s brain activity to model another’s, we can measure shared information across brains—even in dynamic, naturalistic scenarios where an explicit response model may be unobtainable.” (Nastase et al., 2019).

Indeed, in the current dataset, spoken utterances during recall are not necessarily separated by pauses. Instead, they are delineated by the subjective assessment of human raters, who use pauses as just one of the criteria. In addition, the human raters separated utterances based on implied commas. These implied commas often occur in conversational human speech, and while they sometimes involve a pause, they can occur without pauses, such as when a speaker swiftly shifts to a new topic. For example, one participant said: “Watson sits down and Sherlock stands up, and they’re talking / and Watson realizes he just sent a text to the murder / and Sherlock pulls out the pink case and says he found it.” The human raters segmented this sentence into three utterances (shown as slashes), even though the speaker did not pause between these distinct “thoughts.”

We have now expanded the paragraph in the Methods section that describes these recall utterances and how they are defined, so that readers have a more complete understanding of these data. This section, on pg. 23, now reads: “Each participant’s speech from the recall session was transcribed. The transcribed speech was then divided into individual utterances, where each utterance roughly corresponded to a single complete sentence. For long sentences that contained three or more subordinate clauses, sentences were broken up into multiple utterances. Utterance boundaries were subjectively defined as breaks in the participant’s speech. These breaks could occur, for example, during pauses in speech, or during shifts in the recalled content, such as when the speaker switches to a new topic.”

As noted in the Data Availability statement on pg. 29, readers can access the recall transcripts for each participant retrieve the recall transcripts for each participant in the Open Science Framework (OSF) data repository associated with this manuscript at <https://osf.io/8gmes/>. This data release includes transcriptions of each participant’s spoken recall, including the corresponding utterance segmentations and labels for 1) the micro-segments described by each utterance 2) the label of “summary” or “precise” recall for each utterance.

3. The authors concluded “with a plurality of the identified parcels falling within the DMN”. However, for the 7 networks in Schaefer’s parcellation, DMN has more parcels than some other networks. This should be taken into account. Probably should calculate the ratio of the number of identified parcels belong to DMN / the total number of DMN parcels.

The sentence that the reviewer quotes accurately reports that DMN parcels comprise the plurality of the above-threshold parcels in the parcel map shown in Figure 4B. However, as the reviewer rightfully points out, the Figure 4B map includes more parcels from the DMN than from some of

the other networks. To clarify this issue, we calculated the proportion of DMN parcels that show a positive relationship between summarization and memory transformation (i.e., Figure 4B), out of the total number of DMN parcels in which this effect is tested (i.e., parcels included in the mask, which are those that show memory transformation effects, see Figure 3A).

This proportion can be calculated for each network using the data provided in Table 1 to divide row 5 (the number of parcels that show a relationship between memory transformation and summarization) by row 4 (the number of parcels included in the memory transformation mask). These proportions are as follows:

DMN: $11/39 = 28\%$

Fronto-Temporal: $8/21 = 38\%$

Visual: $9/29 = 31\%$

Ventral Attention: $8/18 = 44\%$

Dorsal Attention: $3/15 = 20\%$

Somato-Motor: $2/9 = 22\%$

Limbic: $0/6 = 0\%$

According to this metric, the proportion of the tested DMN parcels that show the positive relationship between summarization and memory transformation is lower than the proportions calculated in the Ventral Attention, Fronto-Temporal, and Visual networks.

To acknowledge this caveat, we have added the following information in the Supplementary Materials as Supplementary Note 2:

“The proportion of tested parcels (i.e., those in the memory transformation mask in Figure 3A of the main text) where memory transformation scaled with summarization varied across networks: 44% of tested parcels in the Ventral Attention Network, 38% of the Fronto-Temporal parcels; 31% of the Visual Network Parcels; 28% of the DMN parcels; 22% of the Somato-Motor Parcels, and 20% of the Dorsal Attention Parcels.”

This note is now cited on pg. 14 of the main manuscript text: “For proportions calculated based on the mask in Figure 3A, see Supplementary Note 2.”

If the reviewer feels strongly that this version of the analysis should be reported in the main results, we would be happy to move it there.

4. The threshold for compression is 10s events. Please give more justification on why 10s was determined? How much would the results change if another temporal threshold was used?

We thank the reviewer for raising this important issue. The ten-second threshold for defining a recall utterance as “summarization” was chosen with the goal of identifying a compression level that would capture the variation in the temporal scale with which people describe past events in our dataset. In the *Sherlock* movie, several actions and details typically unfold in any given ten-second segment of time. In the coarse-grained event boundaries for *Sherlock* (50-scene segmentation, defined in the previously published paper), the average scene duration is 57sec

(min: 11s, max: 180s). If the threshold was set to less than 10 seconds, then by definition it would be impossible for a participant to summarize the shortest movie scene, which is 11s long. However, these choices are qualitative, based on our own judgment given the information available.

For this reason, we created a complementary measure which does not depend on the ten-second cutoff: a continuous measure of temporal compression during recall which we term “temporal compression factor” (TCF). In brief, TCF quantifies the words-per-second that a participant uses to describe each micro-segment of the movie. Fewer recall words per second of movie content lead to higher TCF scores, whereas a higher word count per second translates to a lower TCF score. In the original version of the manuscript, we described this method in the Supplementary Methods, pg. 4., but did not report the parcel maps that show the relationship between TCF and memory transformation.

In our revised manuscript, we now test the correlation between TCF scores and memory transformation (the complement to testing the relationship between recall summarization and memory transformation using the 10-second cutoff, as described in the main manuscript text, Fig. 4B). The resulting parcel map, now shown in Supplementary Figure 3 (pg. 3), is qualitatively very similar to the main map shown in Fig. 4B of the main text. The correlation across the parcel values in the two maps is $r = .65$. In both maps, in parcels in bilateral posterior medial cortex and right lateral temporal cortex, greater temporal compression during recall is associated with greater shifts in encoding-to-recall patterns across people.

Supplementary Figure 3. Parcel-level map of the relationship between memory transformation and scene-level temporal compression factor (TCF) scores (see Supplementary Methods below). This relationship was only tested in parcels that showed reliable memory transformation effects (shown as light gray and colored parcels, see Figure 3A of main text). The parcel values yielded by this analysis are moderately correlated ($r = .65$) with the parcel values reported in the main manuscript text (see Figure 4B).

We now direct the reader to this complementary TCF analysis in the Methods on pg. 24-25: **In contrast to the Direct Judgment method, the Automatic method considers the micro-segment(s)**

assigned to each recall utterance and then sums the duration of these segments to determine whether each utterance's described movie events elapsed in less than or greater than ten seconds. Additional details for the Automatic method, and details for measuring temporal compression as a continuous measure (termed "temporal compression factor", or TCF), are provided in the Supplementary Materials as Supplementary Methods. A parcel-level map of the relationship between TCF and memory transformation are shown in Supplementary Figure 3.

5. Please provide more details of the MRI data acquisition. E.g., what's the TR?

We have added an "MRI acquisition" paragraph to the Methods section on pg. 22:

"Whole-brain anatomical and functional MRI data were collected on a 3 Telsa Siemens Skyra scanner. Anatomical images were acquired with a T1-weighted MPRAGE pulse sequences (0.89 mm³ resolution). T2*-weighted functional images were collected with a gradient echo planar imaging sequences in 27 ascending interleaved slices with 3 x 3 mm² voxels and a 1.5-second TR (echo time: 28 ms, flip angle: 64 degrees, field-of-view: 192 x 192 matrix, slice thickness: 4.0 mm)."

6. Please also provide more details of the data preprocessing for the brain imaging.

We have added an "fMRI preprocessing" paragraph on pg. 22:

"Preprocessing was performed using FSL (<http://fsl.fmrib.ox.ac.uk/fsl>) and included motion correction, slice time correction, linear detrending, high-pass filtering (140 s threshold), and coregistration and affine transformation of the functional volumes to a brain template (MNI, Montreal Neurological Institute, standard). Functional images were resampled to a spatial resolution of 3 mm isotropic voxels. All calculations were performed in volume space. As a final step, projections onto a cortical surface for visualization were performed using the Human Connectome Project (HCP) workbench (<https://www.humanconnectome.org/software/connectome-workbench>)."

7. Table 1 "Results were thresholded at the p<.05 level, except for rows 3-4 (p<.10)." Here "p" should be changed to "q" (FDR)?

The reviewer is correct and the caption for Table 1 has been corrected to state q=.10 instead of q=.05.

8. Line 721 "(283/400 parcels, see Figure 3D)". There was no Figure 3D in the manuscript.

This has been corrected, this line now refers to "Figure 2D" instead of "Figure 3D."

9. Please check the format of the references, some journals use abbreviations and some not, E.g., "Journal of Neuroscience (30)" and J. Neurosci (22).

This has been corrected. All journal names have been abbreviated.

In addition to the revisions that are described above, we have made two additional revisions to ensure that the manuscript follows the journal formatting guidelines:

1. We updated Figure 1A to avoid the use adaptation of previously published images. This figure previously displayed screenshots from the popular television show that was used during the movie viewing phase of fMRI data collection. These images have been replaced with icons that sufficiently illustrate the same information as the previous screenshots. These icons have been purchased so that they do not require attribution. The new version of the figure and caption are shown below:

Figure 1. Experiment design and recall behavior. A) Each subject participated in two fMRI scanning runs. Left: During Run 1, participants watched a 50-minute movie, BBC’s Sherlock, Episode 1, during fMRI scanning. Right: During the immediately following Run 2, participants recalled the movie content via verbal recall. Participants’ responses were audio recorded, transcribed, and segmented into utterances that roughly aligned with ends of sentences and breaks in speech (each row). Each utterance was categorized according to the movie content that it referred to, and according to its temporal precision (i.e., the amount of time that it took for the described events to elapse in the movie). Utterances that described events that occurred in ten seconds or less were coded as “Temporally Precise,” while events that spanned longer than ten seconds were coded as “Summary.” Utterances were coded as “Other” if they did not provide literal and accurate descriptions of the movie events. B) Violin plots depicting the Summary bias across scenes for each individual subject (top row) and across subjects for each individual scene (bottom). Scenes shown in grey were recalled by less than 5/17 participants and were excluded

from all subsequent inter-subject analyses. C) Left: Subject-specific timelines of recall utterances, colored by degree of temporal precision. Right: Proportion of spoken words during recall that were included in utterances coded as Summary, Precise, or Other. D) Diagram of scene durations during movie viewing (y-axis) and movie recall (x-axis) and scene order during recall (diagonal) in a representative participant. Each white box shows one scene from the original 50-scene segmentation. Overlaid on top are the durations and scene identities of the specific recall utterances, coded by temporal precision. The right and left insets show zoomed-in subsets of this recall behavior. For illustration purposes only, “Other” recall utterances are labeled as the most recently described movie scene. The images in Figure 1A were purchased from The Noun Project, <https://thenounproject.com>.

2. We added the following “Statistics and Reproducibility” paragraph to the methods section, as required by the journal. This additional section does not contain any new information that was not included in the previous draft. Rather, in the previous instances where we had separately described the statistical testing procedure for each individual analysis, these descriptions have now been moved here and combined into one section. This section reads as follows (pg. 27-28):

For the text-based analysis comparing the similarity between USE vectors for the movie annotations and spoken recall, significance was evaluated using paired t-tests (two tailed) to compare cosine similarity for vectors of matching segments (i.e., movie segments and recall utterances that covered the same content) versus mismatching segments across all participants (n=17). This was performed separately for each recall type (i.e., movie segments versus temporally precise recall utterances, and movie segments versus summarized recall utterances). To directly compare participants’ average movie-to-recall cosine similarity for Summary versus Temporally Precise recall segments, the cosine similarity values for the matching segments of each recall type were submitted to a paired t-test (two-tailed). These t-tests were deemed significant at $p < .05$ (two-tailed).

For the within-subjects reinstatement analysis in the PMC ROI, the resulting reinstatement values across subjects (n=17) were submitted to a one-sample t-test versus zero (two tailed), computed separately for Summary and Temporally Precise Reactivation. To directly compare reinstatement across recall type, each subject’s two reactivation values were submitted to a paired t-test and deemed significant at $p < .05$ (two-tailed). For the parcel-based analysis, one-tailed p-values were computed at each of the 400 parcels by comparing the group-average reinstatement value to a null distribution that was generated by randomly shuffling the segment-to-utterance correlation matrices 1000 times. To determine thresholding and correct for multiple comparisons, the parcel-level p-values were submitted to False Detection Rate (FDR) correction ($q = 0.05$). Statistical significance and FDR correction were computed separately for the Summary reinstatement and Temporally Precise reinstatement maps.

To compute the parcel map that contrasts Summary versus Temporally Precise within-subject reinstatement, analyses were limited to parcels that showed a reliable Summary or Temporally Precise reinstatement effect (283/400 parcels, see Figure 2D). Subsequent FDR correction was limited to parcels included in this mask. At each parcel in the mask, we directly compared the two reinstatement values across subjects (n=17) with a paired t-test (two-tailed). No parcels survived FDR correction at $q = 0.05$; at a more liberal threshold ($q = 0.10$), ten parcels showed marginal effects of greater reinstatement for Temporally Precise versus Summarized recall.

For the inter-subject pattern similarity analyses (i.e., movie-to-recall and recall-to-recall), statistical significance was determined by shuffling the scene labels 1000 times to generate to a null distribution. For each analysis, a one-tailed p-value was calculated as the proportion of values from the null distribution that were equal to or greater than the observed similarity values for matching scenes. The resulting p-values were then corrected for multiple comparisons using FDR correction ($q=.05$). To test for reliable a correlation between %Summary bias and memory transformation, we submitted each participant's ($n=17$) scene-level correlation between %Summary bias and memory transformation values to a one-sample t-test versus zero (two-tailed). This test was performed at each parcel included in the memory transformation mask, which was limited to parcels with either significant movie-to-recall or significant recall-to-recall similarity. Across parcels in the memory transformation mask, FDR correction ($q=0.05$) was then applied to determine statistical thresholding.

Works Cited

Hasson, Uri, Yuval Nir, Ifat Levy, Galit Fuhrmann, and Rafael Malach. "Intersubject synchronization of cortical activity during natural vision." *Science* 303 (2004): 1634-1640.

Hasson, Uri, Rafael Malach, and David J. Heeger. "Reliability of cortical activity during natural stimulation." *Trends in cognitive sciences* 14 (2010): 40-48.

Nastase, Samuel A., Valeria Gazzola, Uri Hasson, and Christian Keysers. "Measuring shared responses across subjects using intersubject correlation." *Social Cognitive and Affective Neuroscience* 14 (2019): 667-685.

REVIEWERS' COMMENTS:

Reviewer #2 (Remarks to the Author):

The most points I raised in the first-round review have been well addressed. But, with one exception, the measurement reliability remains elusive due to the very limited sample size while the split-half reliability stands around a moderate level. This should be discussed in the final version of the publication to pave the way for the new fMRI protocols regarding the potential challenges and future directions.

Reviewer #3 (Remarks to the Author):

I think the manuscript has been significantly improved since the first version, my concerns have been adequately addressed.